# Exceptional Mechanical Properties and Heat Resistance of Photocurable Bismaleimide Ink for 3D Printing

**DOI:** 10.3390/ma14071708

**Published:** 2021-03-30

**Authors:** Wenqiang Hua, Qilang Lin, Bo Qu, Yanyu Zheng, Xiaoying Liu, Wenjie Li, Xiaojing Zhao, Shaoyun Chen, Dongxian Zhuo

**Affiliations:** 1College of Materials Science and Engineering, Fuzhou University, Fuzhou 350108, China; huawq1996@163.com (W.H.); linqilang@fzu.edu.cn (Q.L.); 2College of Chemical Engineering and Materials Science, Quanzhou Normal University, Quanzhou 362000, China; pkuqubo@163.com (B.Q.); yy410529@163.com (Y.Z.); liuyingzi_1979@163.com (X.L.); wjli2016@163.com (W.L.); zhaoxj199011@163.com (X.Z.)

**Keywords:** 3D printing, bismaleimide resin, mechanical properties, thermal properties

## Abstract

Photosensitive resins used in three-dimensional (3D) printing are characterized by high forming precision and fast processing speed; however, they often possess poor mechanical properties and heat resistance. In this study, we report a photocurable bismaleimide ink with excellent comprehensive performance for stereolithography (SLA) 3D printing. First, the main chain of bismaleimide with an amino group (BDM) was synthesized, and then, the glycidyl methacrylate was grafted to the amino group to obtain the bismaleimide oligomer with an unsaturated double bond. The oligomers were combined with reaction diluents and photo-initiators to form photocurable inks that can be used for SLA 3D printing. The viscosity and curing behavior of the inks were studied, and the mechanical properties and heat resistance were tested. The tensile strength of 3D-printed samples based on BDM inks could reach 72.6 MPa (166% of that of commercial inks), glass transition temperature could reach 155 °C (205% of that of commercial inks), and energy storage modulus was 3625 MPa at 35 °C (327% of that of commercial inks). The maximum values of T_-5%_, T_-50%_, and T_max_ of the 3D samples printed by BDM inks reached 351.5, 449.6, and 451.9 °C, respectively. These photocured BDM inks can be used to produce complex structural components and models with excellent mechanical and thermal properties, such as car parts, building models, and pipes.

## 1. Introduction

In three-dimensional (3D) printing, objects can be fabricated through computer-controlled layer-by-layer accumulation of materials without using molds [1]. The advent of this technology has significantly decreased the production cycle and costs of some industrial products compared to traditional manufacturing methods [2,3,4]. Generally, 3D printing is applied in aerospace technology, medical engineering, construction industry, and electronic manufacturing, among other fields [5,6,7,8,9]. Among the various types of 3D printing technologies, stereolithography (SLA) presents several advantages such as short curing time, high printing accuracy, and energy saving; moreover, the molding materials have excellent performance in terms of hardness, chemical resistance, and abrasion resistance [10,11,12]. However, the development of SLA is restricted by some inevitable shortcomings, particularly in mechanical properties and heat resistance.

With the rapid development of 3D printing technology, higher requirements have been put forward for the quality of light-cured 3D printing products. As printing materials for SLA technology, photosensitive resins mainly include epoxy, acrylic resin, and polyester resin [13,14,15]. In recent years, researchers have improved the comprehensive performance of photosensitive resins through various methods, such as the development of novel resins, adjustment of existing formulations, and addition of inorganic fillers [16,17,18,19,20,21]. For example, Guo et al. synthesized polyimide inks to print products with a tensile strength of 24.9 MPa and decomposition temperature of 432 °C [22]. Borrello et al. reported a polymer resin yielding elastic moduli between 0.6 and 31 MPa, and they obtained these values by simply altering the ratio of monomer and crosslinker feedstocks in the formulation [23]. Asais et al. used calcium sulfate whiskers to modify the hardness and tensile strength of samples (79.3 HD and 30.4 MPa, respectively) [24]. However, the simultaneous improvement in the mechanical properties and heat resistance still remains a challenge. Developing novel photopolymers to widen the application fields of 3D-printed objects is currently a trending topic in materials science.

Bismaleimide (BMI) is a high-performance thermosetting resin with good dielectric properties, high glass transition temperature, and excellent thermal stability, and it is widely used in aerospace, transportation, and machinery electronics, among other fields [25,26,27,28]. At present, the main curing method of BMI is thermal curing, and its biggest disadvantages are the high curing temperature and long production cycle, which easily cause residual stress in the cured product and make it difficult to obtain a highly performing resin [29,30]. Therefore, attaining better curing conditions has become an important aspect of research on BMI modification.

From this viewpoint, we developed a novel bismaleimide resin for SLA 3D printing. The resin based on N,N′-(4,4′-diphenylmethane) bismaleimide–4,4′-diaminodiphenyl methane–glycidyl methacrylate (BDM-DDM-GMA) was prepared via a grafting reaction between BDM and DDM by adding a drop of GMA containing a copolymer inhibitor. The BDM-DDM-GMA presented herein not only effectively solves the shortcomings of the poor moldability of bismaleimide resin but also significantly broadens the application range of light-curing 3D printing resins.

## 2. Materials and Methods

### 2.1. Materials

BDM, DDM, acetate anhydrous, acetone, tetraethylammonium bromide (TEAB), GMA, hydroquinone, 2-hydroxyethyl methacrylate (HEMA), hydroxypropyl methacrylate (HPMA), 2,4,6-trimethyl benzoyl diphenyl phosphine oxide (TPO), 1-benzoylcyclohexanol (Irgacure 184), N,N-dimethylformamide (DMF), tetrahydrofuran (THF), cyclic trimethylopropane formal acrylate (CTFA), trimethylolpropane triacrylate (TMPTA), and isobornyl acrylate (IBOA) were purchased from Aladdin Industrial, Inc. (Shanghai, China) Urethane methacrylate (YYUV-1), polyethylene glycol (200) dimethacrylate (YYUV-2), ethoxylated pentaerythritol tetraacrylate (YYUV-3), polyurethane acrylate (YYUV-4), and aliphatic urethane acrylate (YYUV-5) were supplied by Yongyue Science & Technology Co., LTD (Quanzhou, China).

### 2.2. Synthesis of BDM-DDM-GMA Oligomer

First, DDM (39.6 g, 0.2 mol) was gradually added to a stirred solution containing BDM (35.8 g, 0.1 mol) and 200 mL acetone under a nitrogen flow at 30 °C; then, 1 mL of anhydrous acetate was added to the solution. Then, the temperature was increased to 60 °C and maintained for 2 h. After vacuumizing the system for 30 min, GMA (56.8 g, 0.4 mol), TEAB (0.42 g, 0.002 mol), and hydroquinone (0.22 g, 0.002 mol) were added to the solution, which was then stirred at 60 °C for 8 h to complete the grafting reaction of methacrylate groups. The product was removed while it was hot, and a small amount of residual acetone was removed using a rotary evaporator. The resulting brownish-red viscous product was BDM-DDM-GMA, and the yield was 95%.

### 2.3. Preparation of Photocurable BDM Inks for 3D Printing

#### 2.3.1. Preparation of Photocurable BDM Inks

Photocurable BDM resins were prepared by homogeneously mixing BDM-DDM-GMA with HEMA and HPMA as reactive diluents, YYUV1–5, and photo-initiators TPO and Irgacure 184. Table 1 lists the formulations of the photocurable BDM inks. The new BDM inks are obtained by replacing YYUV-5 and YYUV-4 in commercial inks, and the following part is mainly comparing the mechanical and thermal performance of these two different inks.

#### 2.3.2. 3D Printing using Commercial and BDM Inks

SLA printer (Form 2, Formlabs Inc., Somerville, MA, USA) was employed to produce 3D architectures. The model was designed using the SolidWorks software and subsequently layered with the help of printing software to control the ultraviolet (UV) light curing of the liquid resin; in this way, the photosensitive inks can be layered and printed according to the pre-designed modeling structure. The resulting samples were washed with ethanol and dried under an air flow. Finally, the samples underwent UV light curing for 10 min and, subsequently, thermal curing via the following procedure: 160 °C/1 h + 180 °C/1 h + 200 °C/1 h + 220 °C/2 h.

### 2.4. Characterization Techniques

Fourier transform infrared (FT-IR) spectra were recorded with an attenuated total reflection (ATR) stage on a Thermo Fisher Scientific (USA) Nicolet iS10 spectrometer, at a resolution of 4 cm^−1^, with 16 scans per spectrum. ^1^H nuclear magnetic resonance (NMR) spectra were obtained on a Bruker (Germany) Avance-neo 400 MHz spectrometer using tetramethylsilane (TMS) as an internal standard and deuterochloroform (CDCl_3_) as the solvent. Rheological behaviors of the inks were investigated by a rheometer (Discovery Hybrid Rheometer-2, DHR-2, TA) at 25 °C with a parallel-plate configuration (the diameter and the gap between two geometries were 40 and 0.5 mm, respectively). The range of steady-state shear rate was set from 0.1 to 100 s^−1^. The kinetics of the photocuring process were also investigated with the rheometer equipped with an ultraviolet light emitting diode (UV LED) accessory that had an irradiation power of 80 mW/cm^2^. The gap between two geometries was 0.1 mm. The upper geometry was made of aluminum, and the lower geometry was made of transparent polymethylmethacrylate. The experiment lasted a total of 80 s, and the UV LED light was applied at 20 s. The mechanical properties were evaluated using a universal material testing machine (LD24, Labsans, Shenzhen, China). Specifically, the tensile and flexural behaviors were measured according to International Organization for Standardization (ISO) 527 and ISO 604 standards, respectively. Dynamic mechanical analysis (DMA) was performed using a TA Instruments (USA) DMA Q800 apparatus. DMA tests were carried out from 35 to 300 °C at a heating rate of 3 °C min^−1^ at 1 Hz, and the dimensions of each specimen were rectangular (35.0 mm × 10.0 mm × 3.0 mm). Thermogravimetric analysis (TGA) was performed on a TA Instruments (USA) STA449C machine in the range of 35–800 °C under a nitrogen atmosphere with a flow rate of 100 mL min^−1^ and at a heating rate of 10 °C min^−1^. The surface fracture morphology of the 3D-printed parts was observed by a scanning electron microscope (SEM; JEOL JEM-2010, Tokyo, Japan) at 2.0 kV. Conductivity was improved by adding a thin layer of gold particles on the fracture surface.

## 3. Results and Discussion

### 3.1. Synthesis of the Photosensitive BDM-DDM-GMA Oligomer

The synthesis of the photosensitive BDM oligomer is illustrated in Scheme 1. The carbon–carbon double bond in the BDM molecular chain is influenced by two adjacent electron-withdrawing carbonyl groups, making such bond highly active and prone to undergo Michael addition reactions with compounds containing active hydrogen, such as aromatic amines [31]. The linear oligomer of BDM and DDM was obtained via Michael addition, which extended the main chain of BDM and reduced the number of reactive groups per unit volume, thereby decreasing the crosslinking density of BDM. Because of the destruction of structural integrity of BDM molecules, the solubility of resin in common solvents was improved, and the resin was toughened to a certain extent. Under the action of the catalyst, the epoxy group of GMA reacted with the amino group in the oligomer, thereby forming the required unsaturated double bond.

The chemical structures of the intermediate BDM-DDM and BDM oligomer were confirmed by FT-IR spectroscopy (Figure 1a). The absorption peak relative to the stretching vibration of the primary amine, –NH_2_, can be observed in the BDM-DDM spectrum at 3450–3360 cm^−1^, the peak at 3000–2800 cm^−1^ corresponds to the –CH_2_– stretching vibration, appearing after the imide ring addition, and the peak at 1615 cm^−1^ results from the bending vibration of –NH–; moreover, the peak at 690 cm^−1^ observed in the BDM curve is no longer present in the BDM-DDM spectrum, confirming that the C=C double bond on the imide ring underwent Michael addition by the amino group. At the same time, the peak at approximately 1712 cm^−1^ did not disappear, indicating that the reaction had no effect on the imide group and that BDM-DDM reacted successfully. By comparing the IR spectra of BDM-DDM-GMA and BDM-DDM, a new peak can be observed in the former at 943 cm^−1^, which is due to the methylene group generated after the reaction of the amino group with the epoxy group in GMA. The appearance of this peak indicates the successful grafting of GMA. In addition, the symmetrical stretching vibration peaks of ester bonds at 1294 and 1169 cm^−1^ are strengthened, confirming the successful reaction between GMA and BDM-DDM.

^1^H NMR spectroscopy was employed to characterize the BDM oligomer. As shown in Figure 1b, the peaks with chemical shifts, δ, of 6.98, 6.88, 6.50, and 6.09 ppm represent aromatic protons in DDM, while the two double peaks with δ of 7.14 and 6.91 ppm represent the protons of the benzene ring in BDM. The signal at 5.55 ppm corresponds to the hydroxyl group generated by the reaction of the epoxy group of GMA with the amino group in the BDM-DDM oligomer. The peaks in the range of δ = 4.17–3.95 ppm correspond to the protons in the –CH_2_– and –CH– groups around the hydroxyl group, formed after the ring-opening reaction of the epoxy group. The chemical shifts at 3.95 and 3.71 ppm can be assigned to the protons in the –CH_2_– group between two benzene rings in BDM and DDM, respectively. The chemical shift at 2.91–3.15 ppm corresponds to the protons in the amino group. In summary, the results of FT-IR and ^1^H NMR spectroscopies prove that the BDM-DDM-GMA oligomer was successfully synthesized.

Excellent solubility is essential for BDM-DDM-GMA to be a light-curing 3D printing material. Table 2 lists the solubilities of BDM-DDM-GMA in various organic solvents. These data demonstrate that BDM-DDM-GMA could be easily dissolved in organic polar solvents such as DMF, toluene, and various active diluents, including HEMA, HPMA, CTFA, and IBOA. The good solubility of BDM-DDM-GMA was mainly attributed to the prolongation of the main chain of BDM and the reduction of reaction groups per unit volume, which decreased the crosslinking density of BDM. At the same time, the presence of GMA in the side chain interfered with the ordering of molecules, thus enhancing the solubility.

### 3.2. Rheological Characterization

The rheological properties and curing kinetics of ink are crucial for SLA 3D printing products, since they determine their printing ability, resolution, and appearance [32,33]. The measured viscosity of photocurable BDM inks remained constant at low shear rates, showing Newtonian properties; with a further increase in the shear rate, the viscosity of the inks began to decrease. The decreased viscosity of the inks under a high shear-rate was useful for maintaining good fluidity during the repeated up and down process of the building plate for the SLA 3D printing. These results can be understood considering that the BDM inks mainly form reversible supramolecular structures dominated by intermolecular association. Under a low shear rate, the shear stress was not sufficient to disassemble the association structures, whereas with a further increase in the shear rate, the shear stress increased enough to destroy the supramolecular assemblies, and such process took place at a much faster rate than the reconstruction rate. The plot in Figure 2a shows that for the BDM inks with BDM-DDM-GMA ≤ 15% w/w (BDM-1–3), a decrease in the viscosity can be observed compared to the commercial ink, whereas the viscosity of BDM inks shows a dramatic upturn at the same shear rate when BDM-DDM-GMA increases to 20–30% w/w (BDM-4–6). This fact can be explained by considering that the -OH groups in BDM-DDM-GMA can form hydrogen bonds, thereby increasing the viscosity of the matrix [34].

The UV LED accessory of the rheometer was used to investigate the curing behavior of the inks. The curing process is characterized by free radical polymerization, accompanied by a minor contribution of cationic polymerization. The variations in storage moduli during the curing procedure are shown in Figure 2b. Before the UV LED was turned on (time < 20 s), the storage moduli of all samples, representing the storage modulus of the uncured resin, remained in a low state and almost unchanged. Once the UV light was turned on (time > 20 s), the storage modulus increased markedly, indicating the occurrence of the crosslinking reaction. After UV curing (time ≈ 40 s), the crosslinking reaction was almost complete, and the storage moduli remained high and nearly constant. It is worth noting that the change in slope of the storage modulus could represent the curing speed of the stereolithography resin, namely a higher slope represents a higher curing speed, and vice versa. When the BDM inks contained BDM-DDM-GMA > 20% w/w, the slopes of storage moduli decreased slightly. However, the overall curing performance was excellent and had no impact on 3D printing.

### 3.3. Mechanical Properties of 3D-Printed Samples

The mechanical properties of the samples printed using commercial ink and BDM inks were tested to evaluate the effect of the introduction of BDM-DDM-GMA. Figure 3 displays the results of the stress–strain relationship obtained from tensile (Figure 3a) and flexural (Figure 3b) tests. To analyze the mechanical properties, the tensile strength, elongation at break, flexural strength, and flexural modulus of 3D-printed samples are also provided in Table 3. As shown in Figure 3a, the tensile strengths of all the samples printed by photocurable BDM inks are higher than those of the commercial ink: their values are closely related to the content of BDM-DDM-GMA, and there is an optimum content of BDM-DDM-GMA leading to the maximum tensile strength. Evidently, the tensile strength of products fabricated with BDM inks can be higher than 50 MPa. The tensile strength of the BDM-5 ink was the highest (72.6 MPa), while the lowest tensile strength of the BDM-1 ink was 52.5 MPa. Specifically, the maximum tensile strength of 3D-printed samples fabricated from BDM inks was approximately 1.6 times higher than that of commercial ink (44.3 MPa). This result can be explained by the interactions taking place at the molecular level. The introduction of the benzene ring increased the intermolecular force due to its higher polarity than that of the aliphatic chain, which in turn increased the tensile strength of the 3D-printed samples. Interestingly, the breaking elongation of the UV-cured samples printed by BDM inks ranged from 9% to 14%, which was also higher than that of the samples printed using commercial ink. As the degree of crosslinking of the light-curable resin increased, the elongation at break of the 3D-printed samples also increased.

In addition to the tensile strength, the flexural strength was obtained from flexural tests (Figure 3b). The flexural strength was also enhanced by the addition of BDM-DDM-GMA, and it showed an increase with growing BDM-DDM-GMA content, as shown in Table 3. The flexural strength of the samples printed with BDM-4 and BDM-5 exceeded 115 MPa, and the flexural modulus reached ~3500 MPa. Specifically, the flexural strength and the flexural modulus of BDM-5 are both approximately 3.5 times higher (117.4 and 4468.4 MPa, respectively) than those of the commercial ink.

In addition to the tensile and flexural properties, the impact strength and hardness of 3D-printed products were tested to further investigate the effect of BDM-DDM-GMA. As presented in Figure 4, the impact strength is proportional to the BDM-DDM-GMA content. In general, the trend of notched impact strength corresponds to the elongation at break in tensile strength tests. The values of impact strength for BDM-1 to BDM-6 (Figure 4a) were in the range of 3.0–5.0 kJ m^−2^. The figure shows that for BDM-DDM-GMA content from 0% to 25% w/w, the impact strength increased. As expected, by adding 25% w/w BDM-DDM-GMA, the impact strength reached the optimum value of 4.29 kJ m^−2^, to be compared to the value for the commercial ink, 2.77 kJ m^−2^. With a further increase in BDM-DDM-GMA beyond 25% w/w, the impact strength shifted downward to 3.73 kJ m^−2^. In Figure 4b, a decrease in hardness can be observed for the samples prepared using BDM inks with BDM-DDM-GMA ≤15% w/w, compared to the commercial ink; in contrast, the hardness of products prepared with BDM inks improved when BDM-DDM-GMA increased to 20–25% w/w. Overall, the introduction of BDM-DDM-GMA significantly improved the mechanical properties of 3D-printed samples, especially those obtained from the BDM-5 ink.

To further study the effect of different mass ratios of BDM-DDM-GMA on the properties of samples 3D-printed using BDM inks, their tensile fracture surfaces were observed by SEM, and the results are shown in Figure 5. When the BDM-DDM-GMA content was 5–15% (Figure 5a–c), the fracture surface was smooth and had a few linear cracks, which is a typical appearance of brittle fracture. It can be seen from the SEM images (Figure 5d–f) of BDM-4–6 that the surface became rougher and more wrinkled. This is because the introduction of BDM-DDM-GMA results in an increase in the number of benzene rings, which in turn increases the degree of crosslinking of the light-curable resin. The result suggests an improvement in the mechanical properties of the 3D-printed samples based on BDM inks. However, when the loading of BDM-DDM-GMA reached 30% w/w, the curing of the BDM inks was affected, and the formation of micro-flaws was observed. These results suggest a deterioration of the mechanical properties, including a decrease in tensile strength, flexural strength, impact strength, and hardness.

### 3.4. Thermal Properties of 3D-Printed Samples

In addition to the mechanical properties, the viscoelastic properties of 3D-printed samples prepared with commercial and BDM inks were further studied using a DMA. The storage moduli and loss tangents (tan δ) of the samples as a function of temperature are illustrated in Figure 6a,b, respectively. The corresponding values of the DMA results are also listed in Table 4 to facilitate the discussion. First, the storage modulus shifted upwards to higher values as the BDM-DDM-GMA content increased from 0% to 25%, and then downwards to lower values for BDM-DDM-GMA ≥30%. Let us discuss the points at T = 35 °C: as BDM-DDM-GMA increases, the storage moduli illustrated in Figure 6a shift from 1107.6 to 3625.0 MPa, and then decrease beyond the maximum point (BDM-DDM-GMA = 30% w/w). The storage modulus of the samples based on the BDM-5 ink was 3.3 times that of the samples prepared using commercial ink. Even the lowest storage modulus (1835 MPa for BDM-1) was 1.7 times higher than that of the commercial ink. Such gain in storage modulus is one more indication pointing at an increase in intermolecular forces, and this is likely due to the introduction of the benzene ring of BDM-DDM-GMA because its polarity is greater than that of the aliphatic chain.

Furthermore, the crosslinking density (ν_e_) of the samples was estimated from DMA using the following equation [35,36]:(1)$$\nu_{e}=\frac{E^{\prime }}{3RT}$$
where E’ is the storage modulus in the rubbery state (T_g_ + 50 °C, where T_g_ is the glass transition temperature), R is the gas constant (8.314 J mol^−1^ K^−1^), and T is the absolute temperature at T_g_ + 50 °C. As can be seen from the resulting values presented in Table 4, the samples based on the BDM inks show an increase in the crosslinking density compared to the sample obtained with commercial ink. The highest value was found for the BDM-5 ink (1.82 × 10^2^ mol m^−3^), which is mutually supported by our previous analysis. For higher BDM-DDM-GMA content, the crosslinking density value then decreases (1.57 × 10^2^ mol m^−3^ for BDM-6). One could hypothesize that in these conditions, the benzene ring could hinder the polymerization to some extent and thereby induce a decrease in crosslinking density.

Based on the above characterization of the thermal properties, the T_g_ of the samples was also obtained from the tan δ results. For the samples printed using commercial ink, the value of T_g_ was 75 °C. After introducing BDM-DDM-GMA into BDM inks, the T_g_ exhibited significant improvement. With an increase in the BDM-DDM-GMA content, T_g_ first increased, reaching the highest value (155 °C) at the optimized loading of 25% w/w BDM-DDM-GMA (representing a 106% increase compared to the commercial ink, T_g_ = 75 °C); then, it decreased (see Figure 6b and Table 4). The improved T_g_ could be attributed to two factors. First, the macromolecular chain of BDM-DDM-GMA, which has a rigid benzene group, could continue to extend with the photopolymerization reaction; in this way, a higher concentration of benzene groups would increase the rigidity at the molecular level, thereby restricting the motion of the macromolecule. Second, the higher crosslinking density could lead to a more significant hindrance to chain movement, so that the movement could only occur at higher temperatures. Meanwhile, BDM-5 has narrower and lower intensity peaks in the range of glass transition. This phenomenon may also be ascribed to the increment of crosslinking density which makes the chain movements harder with increasing steric hindrance and chain interactions.

To further assess the thermal properties of the samples printed using the BDM inks, TGA was carried out at a heating rate of 10 °C min^−1^ in nitrogen atmosphere to investigate the effect of BDM-DDM-GMA content on the thermal degradation of the samples. The TGA curves are shown in Figure 7, while Table 5 summarizes the thermal data: the temperatures at 5% and 50% weight loss (T_-5%_ and T_-50%_, respectively), temperature at the maximum degradation (T_max_), and char residual (CR) yield. All samples presented only one stage of thermal degradation. For the samples printed using commercial ink, T_-5%_, T_-50%_, and T_max_ were 287.5, 357.6, and 342.5 °C, respectively. When the BDM-DDM-GMA content increased in the BDM inks, the T_-5%_, T_-50%_, and T_max_ were significantly enhanced. Such an effect became more pronounced at a relatively lower BDM-DDM-GMA content (≤25% w/w). The maximum values of T_-5%_, T_-50%_, and T_max_ were attained by the samples based on the BDM-5 ink, reaching 351.5, 449.6, and 451.9, respectively. The improvement in the thermal properties can be rationalized by considering the motion, breakage, and thermal decomposition of the polymer chains. Because the macromolecular chain of BDM-DDM-GMA, which has a rigid benzene group, increases the rigidity at the molecular level, the macromolecular motion is limited during the process of thermal degradation. In addition, the CR at 800 °C increased from 7.3% to 14.0% with the change in BDM-DDM-GMA content, indicating an improvement in the thermal resistance capacity.

Furthermore, the samples printed with BMD-5 ink and commercial ink were heated in a muffle furnace at different temperatures for 30 min. As shown in Figure 8, BDM-5 did not undergo any evident changes, except for a mild blackening of the surface, which is due to the oxidation of the diluent at high temperatures. In contrast, the commercial resin presented noticeable yellowing and a large number of internal cracks. The samples prepared using BDM inks exhibited much better thermal stability, which will greatly expand its applications in areas such as automotive parts, high-temperature molds, and aviation.

We collated some data on the mechanical and thermal properties of 3D-printed samples using different photosensitive inks, as shown in Table 6. We observed that the tensile strength of the commercially available photosensitive resins is far from 70 MPa: despite having good thermal resistance, they do not possess exceptional mechanical strength. In contrast, the BDM inks developed in our work, containing different amounts of BDM-DDM-GMA, have outstanding mechanical and thermal properties. Although excellent toughness is essential, good heat resistance and mechanical strength are important factors to consider when using workpieces in demanding applications such as automotive and aerospace technologies. Considering the overall performance of the final products, BDM-5 was the best choice for optimal comprehensiveness.

### 3.5. Fabrication of Objects with SLA 3D Printer

To further assess the properties of the BDM inks in practical applications, some mechanical elements and models including rockets, towers, and gears were fabricated by deposition of a 0.1 mm thick layer using a 3D printer, as shown in Figure 9. The rocket model consisted of three parts, as shown in Figure 9a, and each part could be accurately pieced together into a complete model. All the resultant materials exhibited good surface performance and excellent layer-to-layer binding, which indicates that the 3D printing BDM inks are promising in the field of advanced additive manufacturing. It can be concluded that the addition of BDM-DDM-GMA did not adversely influence the printing accuracy. Remarkably, this means that the BDM-DDM-GMA-enhanced BDM inks can be fabricated without further special optimization for the 3D printer. The favorable compatibility with current commercial 3D printers could further extend the application to fields demanding materials with exceptional mechanical and thermal properties.

## 4. Conclusions

In this study, a novel photocurable BDM resin was synthesized from BDM, DDM, and GMA. Its excellent solubility, good fluidity, and UV-curing performance provide an excellent basis for its application in 3D printing. BDM-DDM-GMA can be prepared with other acrylic resins such as oligomers, HEMA and HPMA as reaction diluents, and TPO and Irgacure 184 as photo-initiators to produce BDM inks for SLA 3D printing. BDM inks can be employed to print parts and models with complex structures, and they exhibit excellent mechanical and thermal properties. In particular, 3D samples printed using the BDM-5 ink possessed a tensile strength of up to 72.6 MPa (166% of that of commercial inks), T_g_ of 155 °C (205% of that of commercial inks), and energy storage modulus of 3625 MPa (327% of that of commercial inks). The maximum values of T_-5%_, T_-50%_, and T_max_ of the samples printed using BDM-5 reached 351.5, 449.6, and 451.9 °C, respectively. We believe that the SLA-based 3D printing capability will be enhanced significantly owing to these high-performance photocurable BDM inks. We foresee the application of these photocurable BDM inks to produce complex structural components and models with excellent mechanical and thermal properties, such as car parts, building models, and pipes.

## Data Availability

Data sharing is not applicable to this article.

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
