# Peer review of "Exceptional Mechanical Properties and Heat Resistance of Photocurable Bismaleimide Ink for 3D Printing"

_materials, 2021, doi:10.3390/ma14071708_

Round 1
Reviewer 1 Report
Overall, the paper is well written and organized. The topic of this study seem to be of interest for a small group of researchers engaged in resin printing. Therefore, authors should have a solid explanation of why this method is important on a broader impact. Discussion section is very small and does not substitute Conclusions. Discussion can be merged with results or appear as a stand-alone section.
The idea of developing customized resins with pre-defined properties of the material is challenging, but feasible. Please focus on wider implications the results obtained.
Thank you!
Author Response
March 13, 2021
Dear Reviewer,
I am sending here with the revised manuscript (materials-1139258) of the paper entitled “Exceptional Mechanical Properties and Heat Resistance of Photocurable Bismaleimide Ink for 3D Printing”. The revised manuscript consists of 15 pages of text. In the revised version, we have fully taken into account all the points made by the reviewer. All changes have been marked in RED color as shown in the revised version.
Thanks very much for your valuable suggestions. I have checked editorial work needed on the manuscript and reconsidered the conclusions. All of the responses are given as follows:
Again, we are very grateful to you for your time and valuable suggestions. I cherish the revised manuscript could be reconsidered and accepted for the publication in Materials.
Very sincerely yours,
Dr. Dongxian Zhuo
College of Chemical Engineering and Materials Science
Quanzhou Normal University, Quanzhou, Fujian, 362000, P. R. China.
E-mail: dxzhuo@qztc.edu.cn

Reviewer 2 Report
Authors of the manuscript entitled “Exceptional Mechanical Properties and Heat Resistance of Photocurable Bismaleimide Ink for 3D Printing” present an original work of preparing a novel photocurable BDM resin for its application in 3D printing. The manuscript covers one current and interesting topic related to 3D printing technologies.
The manuscript is well organized; introduction provide sufficient background and include all relevant references for the topic. Research design is appropriate and well explained. Results are clearly presented and conclusions as well.
The acceptance of the manuscript is suggested with one minor correction. The title Conclusion is missing. The main conclusions are listed in the title Discussion, so I recommend that this last title be renamed the Conclusion. I suggest that section 3. Results is renamed at Results and Discussion, which they are in terms of content.
Author Response
Reviewer 2
March 13, 2021
Dear Reviewer,
I am sending here with the revised manuscript (materials-1139258) of the paper entitled “Exceptional Mechanical Properties and Heat Resistance of Photocurable Bismaleimide Ink for 3D Printing”. The revised manuscript consists of 15 pages of text. In the revised version, we have fully taken into account all the points made by the reviewer. All changes have been marked in RED color as shown in the revised version.
Thanks very much for your valuable suggestions. I have checked editorial work needed on the manuscript and reconsidered the conclusions. All of the responses are given as follows:
Again, we are very grateful to you for your time and valuable suggestions. I cherish the revised manuscript could be reconsidered and accepted for the publication in Materials.
Very sincerely yours,
Dr. Dongxian Zhuo
College of Chemical Engineering and Materials Science
Quanzhou Normal University, Quanzhou, Fujian, 362000, P. R. China.
E-mail: dxzhuo@qztc.edu.cn

Reviewer 3 Report
This paper entitled “Exceptional Mechanical Properties and Heat Resistance of Photocurable Bismaleimide Ink for 3D Printing” presents a novel photocurable BDM resin to be used in stereolithography (SLA), a promising type of 3D printing technology. In this work, some constituents of a commercial ink have been replaced by this novel resin substantially improving both thermal and mechanical properties. This topic is of great interest since SLA is a particularly promising technique because of its short processing times and printing accuracy so all advances made to improve the properties of resins are welcome.
The manuscript is well-written, well structured, and properly documented. The results are clear and reasonably well discussed. Nevertheless, I would like to comment on some minor aspects:
1.-In the Materials and Methods section, I feel there has been a mistake in the numbering of the headings or, if not, that the previous paragraphs have been accidentally deleted. Please, check it.
I also consider that a lot of information is missing. There are numerous compounds or substances named by acronyms whose description has not been included above. For example: TEAB (line 76) , HEMA, HPMA, YYUV1-5, TPO (section 2.3.1)... I know they are "known" compounds to experts in the field, but it is strictly necessary to include their names in order to understand the results of the work.
Just for information, as there are numerous commercially available resins, which commercial resin is being used?
It is assumed that the values shown in Table 1 are percentages by weight. Please include this clarification in the text or the table.
Also, from Table 1, I can guess that the authors are replacing part of two substances of this commercial resin, namely YYUV-5 and YYUV-4, for the new bismaleimide resin BDM-DDM-GMA, right? In my opinion, it is an aspect that is not properly reflected in the manuscript. Please, indicate this point more clearly, as it seems that a new resin is being formulated, and what is really being carried out is a "modification" of a commercial resin.
In 2.3.2. section, the last three lines (lines 98-100) indicate that the samples are being post-treated by UV and heat. Is it real or is it a mistake? As far as I know, it is not usually necessary to apply to the photocured samples by SLA any post-treatment.
In 2.4 section, characterization techniques¸ I miss information about the procedures in some of the characterization tests carried out. Could you give more details?
- In the rheological characterization, which geometry has been used? (plate-plate, cone-plate, concentric cylinders…)
- In the DMA, which deformation mode has been used? What were the dimensions of the samples, were they printed directly or obtained from a larger one?
- As far as mechanical properties are concerned, what type of specimens were used? How many tests were carried out per sample? The standard deviation is not indicated in the results. Please, include it as it is an important aspect to evaluate the obtained results.
2.-In the Results section:
In 3.2. section, Rheological characterization, rheological properties are usually represented on a logarithmic scale. I would appreciate it if you change the scale of the Y (viscosity) and X (shear rate) axes. On the other hand, I consider it very risky to talk about shear thinning in these results. In general, these resins, due to their nature and molecular weight, usually maintain a Newtonian behaviour in the shear rate range considered. It would be advisable to reconsider this discussion with the results plotted on a suitable scale.
When it is used the UV LED accessory of the rheometer, the completion of the crosslinking reaction is noticed by the constancy of the storage moduli. In this case, the storage moduli of the BDM inks are lower than that of the commercial ink, but moduli remain constant so, in my opinion, there is no indication that a post-treatment is needed. As I previously mentioned, the information on the substances that are being substituted is not available, so frankly, it is sort of difficult to make a hypothesis in order to explain these results. However, it is clear that the formulations are different, so the results may also be different. On the other hand, if a post-treatment has been done, have these moduli been characterised to confirm the authors' hypothesis?
In the 3.3 section, Mechanical properties, it would be advisable to indicate the error range in Table 3.
In the 3.4. section, Thermal properties, there are a few points I would like to mention:
- It would be advisable to give the Tg without decimals, especially because the Tg value has been supposedly calculated at the Tan d maximum, which is surprisingly wide in these samples, and because the number of repetitions carried out has not been mentioned.
- Line 283-288. In my opinion, the higher modulus of BDM-5 would be more related to its higher crosslink density and higher and narrower Tg, so that at 35ºC it is normal for it to have a higher value of E'. I am not sure if the polarity of the benzene ring affects this result.
- The Tg of the photocured BDM resins is surprisingly wide, almost a transition of 180°C. However, in most of the formulations, a shoulder corresponding to the structures formed by the compounds being substituted in the commercial one and coinciding with the Tg of the commercial formulation is visible. Logically, this shoulder is hardly noticeable in BDM-5, but it is in BDM-6. I wonder if you could have any hypotheses about these observations.
- The thermal stability results are really fabulous. The improvement in thermal stability achieved with the introduction of a small fraction of BDM-DDM-GMA is remarkable. In my opinion, the results that provide real information are T5% and CR(%). T50 and Tmax, which is not clearly described how they were obtained, are dispensable.
Author Response
March 13, 2021
Dear Reviewer,
I am sending here with the revised manuscript (materials-1139258) of the paper entitled “Exceptional Mechanical Properties and Heat Resistance of Photocurable Bismaleimide Ink for 3D Printing”. The revised manuscript consists of 15 pages of text. In the revised version, we have fully taken into account all the points made by the reviewer. All changes have been marked in RED color as shown in the revised version.
Thanks very much for your valuable suggestions. I have checked editorial work needed on the manuscript and reconsidered the conclusions. All of the responses are given as follows:
2.-In the Results and Discussion section:
In the 3.3 section, Mechanical properties, the error range has been supplemented in Table 3.
Again, we are very grateful to you for your time and valuable suggestions. I cherish the revised manuscript could be reconsidered and accepted for the publication in Materials.
Very sincerely yours,
Dr. Dongxian Zhuo
College of Chemical Engineering and Materials Science
Quanzhou Normal University, Quanzhou, Fujian, 362000, P. R. China.
E-mail: dxzhuo@qztc.edu.cn

Reviewer 4 Report
The article : Exceptional Mechanical Properties and Heat Resistance of
Photocurable Bismaleimide Ink for 3D Printing present very interesting results in a field of the future, that of 3D printing.
Few minor suggestions:
L110: what are the DMA sample dimensions?
L115: what electron gun voltage was used for SEM images was it at low or high vacuum? any special conditions
L122: present a reference for Michael addition reactions (or some explanations)
L291: better quality of the equation + number
L292 is E or E prime
L382: Discussion and conclusions or maybe better only conclusions
L366: provide few 3D printing parameters like rate, layers, thickness, degree of filling etc.
Author Response
March 13, 2021
Dear Reviewer,
I am sending here with the revised manuscript (materials-1139258) of the paper entitled “Exceptional Mechanical Properties and Heat Resistance of Photocurable Bismaleimide Ink for 3D Printing”. The revised manuscript consists of 15 pages of text. In the revised version, we have fully taken into account all the points made by the reviewer. All changes have been marked in RED color as shown in the revised version.
Thanks very much for your valuable suggestions. I have checked editorial work needed on the manuscript and reconsidered the conclusions. All of the responses are given as follows:
- What are the DMA sample dimensions?
Response: The dimensions of each specimen were rectangular (35.0 mm × 10.0 mm × 3.0 mm).
- What electron gun voltage was used for SEM images was it at low or high vacuum? any special conditions .
Response: The surface fracture morphology of the 3D-printed parts was observed by a scanning electron microscope (SEM; JEOL JEM-2010, Japan) at 2.0 kV. Conductivity was improved by adding a thin layer of gold particles on fracture surface.
- Present a reference for Michael addition reactions (or some explanations) .
Response: We have added a reference “Li, Q.; Hu, C.; Ye, W.; Gu, H.; Huang, P. Study on Properties of Phenol Type Epoxy Resin Modified by Bismaleimide. Journal of Aeronautical Materials 2008, 28, 82-86.” (see page 4, line 140). This paper describes in detail the mechanism of chain extension of bismaleimide by Markel addition reaction.
- L292 is E or E prime.
Response: E prime.
- Provide few 3D printing parameters like rate, layers, thickness, degree of filling etc.
Response: Some mechanical elements and models including rockets, towers, and gears were fabricated by deposition of layer with 0.1 mm thick using a 3D printer.
Again, we are very grateful to you for your time and valuable suggestions. I cherish the revised manuscript could be reconsidered and accepted for the publication in Materials.
Very sincerely yours,
Dr. Dongxian Zhuo
College of Chemical Engineering and Materials Science
Quanzhou Normal University, Quanzhou, Fujian, 362000, P. R. China.
E-mail: dxzhuo@qztc.edu.cn
